# pH-Sensitive Folic Acid Conjugated Alginate Nanoparticle for Induction of Cancer-Specific Fluorescence Imaging

**DOI:** 10.3390/pharmaceutics12060537

**Published:** 2020-06-11

**Authors:** Sara Lee, Kangwon Lee

**Affiliations:** Department of Transdisciplinary Studies, Graduate School of Convergence Science and Technology, Seoul National University, Seoul 08826, Korea; saralee@snu.ac.kr

**Keywords:** alginate, folic acid, 5-aminolevulinic acid, nanoparticle, nanomedicine, bioimaging, cancer diagnosis

## Abstract

In cancer nanomedicine, numerous studies have been conducted on the surface modification and transport capacity of nanoparticles (NPs); however, biological barriers, such as enzymatic degradation or non-specific delivery during circulation, remain to be cleared. Herein, we developed pH-sensitive NPs that degrade in an acidic environment and release 5-aminolevulinic acid (5ALA) to the target site. NPs were prepared by conjugating alginate with folic acid, followed by encapsulation of 5ALA through a water-in-oil (W/O) emulsion method. The alginate-conjugated folic acid nanoparticles (AF NPs) were homogeneous in size, stable for a long time in aqueous suspension without aggregation, and non-toxic. AF NPs were small enough to efficiently infiltrate tumors (<50 nm) and were specifically internalized by cancer cells through receptor-mediated endocytosis. After the intracellular absorption of NPs, alginate was deprotonated in the lysosomes and released 5ALA, which was converted to protoporphyrin IX (PpIX) through mitochondrial heme synthesis. Our study outcomes demonstrated that AF NPs were not degraded by enzymes or other external factors before reaching cancer cells, and fluorescent precursors were specifically and accurately delivered to cancer cells to generate fluorescence.

## 1. Introduction

Targeted drug delivery to tumors using nanoparticles (NPs) is an emerging area in cancer nanomedicine. Thus, studies are needed to improve drug selectivity, prolong blood circulation, and reduce the side effects of therapeutic NPs [1,2]. After the discovery of the enhanced permeability and retention (EPR) effect, extensive research into the application of NPs for chemotherapy of solid tumors has begun [3]. Numerous types of NPs have been developed over the years, which are now widely utilized in bioimaging due to their optimized surface modification and the ability to transport various therapeutic components [4]. Fluorescence imaging is a noninvasive bioimaging modality used for cancer diagnosis and as a guide for cancer resection surgery [5,6,7,8]. Over the past few decades, fluorescence-mediated cancer detection has evolved to accurately differentiate cancer tissues, rather than distinguishing cancer tissues based on subjective evaluation with the naked eye [9]. These advances reduce the time and cost of cancer treatment and improve quality. However, despite practical research efforts, biological barriers to the transport of fluorescent materials, such as enzymatic degradation, inadequate accumulation, and nonspecific distribution still remain to be crossed [1,3].

Alginate is a natural anionic polysaccharide, extracted from brown algae. It is widely used in numerous pharmaceutical and biotechnological application due to its biocompatibility, biodegradability, non-toxicity, low cost, and easily controlled physical properties [10]. The general biomedical applications of alginate hydrogels are in drug delivery, cell transplantation, and wound healing [11,12]. In the acidic environment, alginate undergoes acid-catalyzed hydrolysis and these properties are applied to drug release systems [10,13,14]. The properties of alginate can be optimized for specific applications through chemical modification [13]. Folic acid has been extensively studied as a cancer uptake ligand [15,16,17,18]. It is often used as an agent for targeted drug delivery because of its high affinity for folic acid receptors, low cost, non-immunogenicity, and biostability. Folic acid receptors are overexpressed in cancer cell epithelium, which is useful for the targeting of cancer cell membranes and facilitates nanoparticle endocytosis [19,20,21,22]. 5-aminolevulinic acid (5ALA) is a precursor of protoporphyrin IX (PpIX), a photosensitizer that is generated during heme synthesis and accumulates in cancer cells. 5ALA is approved by the U.S. FDA, has few side effects, and has the advantage of being quickly excreted from the body [6,23,24,25], therefore, it is used as a fluorescent agent for cancer diagnosis and surgical resection in many preclinical and clinical applications [7,8]. Generally, 5ALA is transported to the cancer cells by a nanocarrier because of its low selectivity for cancer cells and poor cell permeability, which is engulfed by bacteria [24,26]. Therefore, NPs encapsulated with 5ALA are stable in the circulatory system, but are still required to be safely and accurately delivered to the target cancer cells. Additionally, there are many biopolymer drug delivery systems conjugated with folic acid. Among them, there is a study in which the 5ALA encapsulation ratio is increased by adding alginate [27]. We reduced the cost and conjugated folic acid to alginate in a simple way to increase the 5ALA encapsulation ratio and devised a drug delivery system that could be used in vivo later.

In this research, we fabricated stable NPs to improve selectivity and prolong blood circulation time and pH-dependent degradability to deliver 5ALA to a larger number of cancer cells. Alginate and folic acid were used to improve the cellular uptake efficiency. The carboxyl group of the alginate interacted electrostatically with 5ALA, and the hydroxyl group was conjugated to folic acid. The synthesized NPs reached cancer tissue via blood circulation and bound to folic acid receptors in the cancer cell epithelium, causing their cellular internalization. The release process of 5ALA is shown in Figure 1. AF NPs are bound to folic acid receptors overexpressed on the surface of cancer cells, endocytosis, and then degraded in a lysosomal pH environment, resulting in 5ALA release. The emitted 5ALA is converted to PpIX by heme synthesis, and when the accumulated PpIX is excited at a wavelength of 405 nm, it emits fluorescence at a wavelength of 635 nm. The stability of NPs was demonstrated through physicochemical characterization, and the toxicity assay confirmed that the NPs were not cytotoxic. Therefore, this study resulted in the improved internalization of a cancer imaging agent and proposes a new carrier in which 5ALA is specifically released in tumors in a lysosomal pH-dependent manner.

## 2. Materials and Methods 

### 2.1. Materials

Sodium alginate was obtained at Wako Pure Chemical Industries (11 kDa, Osaka, Japan). Folic acid, tetrabutylammonium hydroxide solution (TBAOH), tetrabutylammonium fluoride hydrate (TBAF), 1,1′-carbonyldiimidazole (CDI), soybean oil, 5-aminolevulinic acid hydrochloride (5ALA, 98%), sorbitan oleate (Span 80), and Polysorbate 80 (Tween 80) were supplied from Sigma-Aldrich (St Louis, MO, USA). Hydrogen peroxide (H_2_O_2_, 30.0–35.5%), dimethyl sulfoxide (DMSO, 99.8%), sodium carbonate (Na_2_CO_3_), and hydrochloric acid (HCl, 35.0–37.0%) were purchased from Samchun (Gyeonggi-do, Korea). 2,4,6-trinitrobenzene sulfonic acid (TNBSA, 5% *w*/*v*) solution, and bicinchoninic acid (BCA) assay kit were obtained from Thermo Fisher (Waltham, MA, USA). The MCF-7 (breast adenocarcinoma), A549 (lung carcinoma), and SKOV-3 (ovarian adenocarcinoma) cell lines were procured from Korea Cell Line Bank (Seoul, Korea). The human fibroblast (HFB) cell line was offered by Seoul National University Hospital (Bundang, Gyeonggi-do, Korea). RIPA lysis buffer was purchased from Biosesang (Gyeonggi-do, Korea) and Cell Counting Kit-8 (CCK-8) was acquired by Dojindo Molecular Technologies, Inc. (Kumamoto, Japan). 

### 2.2. Synthesis of AF 

To synthesize alginate-conjugated folic acid (AF), sodium alginate (1 g, 4.632 mmol) was added to a 30 mL mixture of ethanol/0.6 M HCl and stirred for 18 h at 4 °C. Alginic acid was produced and it was filtered in vacuo using qualitative filter paper (Whatman™), washed several times with alcohol and acetone, transferred to a vial, and dried under vacuum overnight. The dried alginic acid was dissolved in water (30 mL) and 4% TBAOH was added with continuous stirring until the solution reached pH 9. The opaque solution was lyophilized to yield white TBA-alginate.

Folic acid (0.408 g, 0.9264 mmol) mixed with DMSO (10 mL) and CDI (0.1502 g, 0.9264 mmol) was added. The compound was stirred under N_2_ gas at 25 °C in the dark for 24 h. Then, TBA-alginate was dissolved in 50 mL DMSO with 1 wt% TBAF. Under continuous stirring, the folic acid/CDI compound was added to the TBA-alginate solution and left to react overnight in the dark at 40 °C. The product was precipitated in ice cold ethanol/methanol (1:1) containing 0.01 M HCl, filtered, and washed with alcohol. The product was neutralized by dissolving in a solution of sodium carbonate, and AF was obtained by lyophilization. The synthesis process was shown in Figure 2. The structure of AF was analyzed by Proton nuclear magnetic resonance (^1^H NMR) spectra, Fourier transform infrared (FT-IR) spectra, and UV-vis spectroscopy. 

### 2.3. Preparation of 5ALA Loaded NPs

AF NPs were fabricated using a W/O emulsion method with probe sonication [28]. Briefly, soybean oil, co-surfactant (a mixture of Span80 and Tween80), and the aqueous phase (AF, 5ALA) were mixed in a glass vial at a weight ratio of 7:2:1. For particle optimization, different NP samples were prepared by adjusting the hydrophilic-lipophilic balance (HLB) values and AF concentration (0.5–1 wt%), while the concentration of 5ALA was fixed at 1 wt%. The solution was mixed, followed by sonication without on-off pulse using a probe-type sonicator (VC-750, Sonics and Materials, Newtown, CT, USA) for 10 min at 40% amplitude. Then, the mixture was obtained as a yellowish opaque solution, which was redispersed in deionized (DI) water or Phosphate-buffered saline (PBS) and centrifuged. The NPs gathered at the liquid interface were collected and filtered using a syringe membrane filter (DISMIC-25, Advantec, Tokyo, Japan). Lastly, dialysis was performed for 24 h using a dialysis membrane (Cellu-Sep MWCO 25 kDa) to remove impurities from the NP solution

### 2.4. Characterization

^1^H NMR spectra (JNM-LA400, JEOL, Tokyo, Japan) were obtained at 400 MHz; alginate and AF were measured at 80 °C and other compounds were measured at 25 °C. FT-IR spectra (ALPHA, Bruker, Billerica, MA, USA) were recorded at frequency range 4000–400 cm^−1^ to characterize AF. UV-vis spectral analysis was performed by a Nanodrop 2000 spectrophotometer (Thermo Fisher, USA).

The average size, distribution frequency by size, and ζ potential of AF NPs were analyzed using a Zetasizer (Malvern, UK). The scattering was measured at 25 °C at an angle of 173°. The NPs were stored at 4 °C and stability was assessed by recording the NP size over a period of three months. Transmission electron microscopy (TEM JEM-3010, JEOL, Tokyo, Japan) was used to observe the morphology of AF NPs.

### 2.5. Encapsulation Efficiency and 5ALA Release Profiles 

The concentration of 5ALA contained in the AF NPs was measured by NP decomposition. AF NPs (1 mL) were dispersed in 1.5% hydrogen peroxide (1.5 mL) and sonicated in an ultrasonic water bath for 10 min at 37 °C, then stirred vigorously for 2 h. After NP decomposition was complete, 5ALA was separated by centrifugation at 12,300× *g* using a Microsep device (MWCO 1 kDa). The suspension was collected, lyophilized, and 5ALA quantified using TNBSA solution according to the manufacturer’s instructions. The 5ALA loading capacity (LC) and encapsulation efficiency (EE) in AF NPs was calculated as per the equations presented:$$\mathrm{LC}\left(\%\right)=\frac{Entrapped\;amount\;of\;5ALA}{Nanoparticles\;weight}\times 100$$
$$\mathrm{EE}\left(\%\right)=\frac{Entrapped\;amount\;of\;5ALA}{Total\;amount\;of\;5ALA}\times 100\;$$

The release profile of 5ALA was assessed under two different pH environments (PBS at pH 5.0 and pH 7.4) at 37 °C. 5ALA-containing AF NPs (2 mL) were placed in a dialysis membrane (MWCO 1 kDa) in 8 mL PBS at pH 5.0 or pH 7.4 and stirred at 37 °C, to simulate the lysosomal and normal physiological pH values, respectively. After a fixed time, the release solution taken out and fresh PBS was added. The concentration of 5ALA in the dialysis solution was analyzed on a microplate reader (Synergy H1, BioTek, Winooski, VT, USA) using TNBSA solution according to the manufacturer’s instruction. The cumulative 5ALA concentration released from AF NPs was calculated based on the following equation:$$\mathrm{Cumulative}\;\mathrm{release}\;(\%)=\frac{V_{0}C_{s}+V_{r}\sum C_{s-1}}{T_{ALA}}\times 100$$
where $T_{ALA}\;$ indicates the total amount of 5ALA in the AF NPs, *V*_0_ is the total volume of the release solution (*V*_0_ = 10 mL), *V_r_* is the volume of the added PBS (*V_r_* = 1 mL), and *C_s_* indicates the concentration of 5ALA in the sample.

### 2.6. Cell Culture

The human fibroblast cell line HFB was cultured in Dulbecco’s Modified Eagle’s Medium (DMEM) with 10% fetal bovine serum (FBS) and 1% penicillin-streptomycin (PS). MCF-7, A549, and SKOV-3 cell lines were cultured in RPMI 1640 with 10% FBS and 1% PS. All cell lines were cultured in a humidified 37 °C constant temperature incubator set to 5% CO_2_.

### 2.7. Quantification of PpIX

To quantify the intracellular accumulation of PpIX, cells were placed at 0.5 × 10^6^ cells per well in 24-well plates. The medium cultured for 48 h was replaced with fresh serum-free media containing AF NPs (0.1 mg/mL) and incubated for 24 h. After uptake of AF NPs and conversion of 5ALA to PpIX, the culture media containing NPs was sucked off and cells were washed by PBS. Then, 100 µL ice-cold RIPA lysis buffer was treated to each well to extract PpIX, mixed well, and incubated on ice for 30 min, vortexing 4–6 times. At 4 °C for 20 min, the lysis solution was centrifuged at 14,000× *g*. The supernatant was placed to a black 96-well plate and then measured with a microplate reader for fluorescence intensity analysis at 635 nm emission wavelength (405 nm excitation wavelength) and fluorescence gain 150. BCA assay was used to obtain a quantitative fluorescence value according to the cell numbers by normalizing the fluorescence intensity to the total protein concentration of the lysate.

### 2.8. Cytotoxicity

For the cytotoxicity test, cells were placed at 5 × 10^3^ cells per well in 96-well plates and preincubated for 24 h at 37 °C. The four cell lines were incubated with different concentrations of NPs (12.5, 25, 50, 100 or 200 μg/mL) at 37 °C for 6, 12 or 24 h. After removing the NP-containing media, CCK-8 solution in media was treated and reacted in a 37 °C incubator for 2 h. Absorbance was observed at 450 nm using a microplate reader.

### 2.9. Cellular Internalization and Fluorescence Image of PpIX Generation

To assess the cellular uptake of AF NPs, all cell lines were placed at 0.05 × 10^6^ cells per well in 48-well plates and preincubated at 37 °C for 48 h. The media was replaced with fresh media without serum containing AF NPs (0.1 mg/mL) and incubated for 12 or 24 h. The cells were washed by PBS and added 100 μL 4% paraformaldehyde solution to each well to fix for 15 min. 4′,6-diamidino-2-phenylindole (DAPI) was used for cell nuclei staining and placed at room temperature for 20 min. After the staining procedure, cells were washed using Dulbecco’s PBS and fluorescence image was observed through a confocal fluorescence microscope (Zeiss Z1 Axio Observer, Carl Zeiss, Oberkochen, Germany). Fluorescence of PpIX was observed under the conditions of AT560/40 nm excitation and 635/60 nm emission filters.

## 3. Results and Discussion

### 3.1. Synthesis and Characterization of AF 

Alginate is soluble only in aqueous solutions. To allow sodium alginate to be dissolved in an organic solvent, it was modified with TBAOH [29,30]. The TBA-alginate was characterized by FT-IR and ^1^H NMR spectroscopy and compared to the spectra of unmodified alginate. In the FT-IR spectrum, O-H stretching is observed at approximately 3400 cm^−1^. A new peak corresponding to the aliphatic C-H stretching of TBA was seen at 2870–2960 cm^−1^. The characteristic ^1^H NMR peaks occurred at 3.5 ppm (-CH_2_-), 2 ppm (-CH_2_-), 1.7 ppm (-CH_2_-), and 1.3 ppm (-CH_3_) (Appendix A).

AF (600.52 Da) was successfully conjugated by esterification. Esterification occurs in the carboxylic acid portion of secondary carbon, which is more reactive in folic acid. In addition, the CDI of the primary carbon carboxylic acid portion of folic acid is removed by treating the strong acid (HCl) in the precipitation process after the reaction [31,32]. The synthesis of AF was confirmed by ^1^H NMR. The ^1^H NMR spectra represents alginate, folic acid, and AF (Figure 3). In the ^1^H NMR spectra of AF, the peaks at 8.1 and 7.3 ppm were attributed to the aromatic protons of folic acid. Furthermore, the peaks at 7.6, 8.2, and 9.1 ppm corresponded to protons of the pterin ring of folic acid, and the peaks appearing from 4 to 5.5 ppm were peaks associated with the basic structure of alginate. Compared to the folic acid ^1^H NMR spectrum, the peaks corresponding to folic acid in the AF spectrum were shifted slightly. These results indicate that folic acid was successfully conjugated to alginate.

As shown in Figure 4, the FT-IR spectrum of AF revealed ester carbonyl stretching vibrations (C=O) at 1727cm^−1^ and C-O-C stretching vibrations at 1378 and 1188 cm^−1^. The folic acid reference spectrum showed an absorption peak between 3600 and 3280 cm^−1^ because of –OH and NH–, respectively. The peaks observed at 1685 and 1602 cm^−1^ corresponded to the C=O and N–H of the carboxyl group, respectively. The specific peak at 1480 cm^−1^ was attributed to the phenyl ring [33]. Sodium alginate showed a specific peak at 3317 cm^−1^ as a result of the hydroxyl group (–OH), and at 1600 cm^−1^ as a result of asymmetric stretching of COO bonding. Another specific peak shows at 1406 cm^−1^ as a cause of symmetric vibration. Additionally, the peak at 1022 cm^−1^ indicated elongation of C-O [34,35]. All peaks corresponding to folic acid and alginate were identified in the AF IR spectra.

To confirm that folic acid was covalently bound to alginate, UV-vis spectroscopy was carried out. The AF spectrum showed absorption at 275 nm (Figure 4). Folic acid absorbed light at approximately 285 nm, which was due to the typical π→π^*^ transition of the pterin ring. In addition, AF showed absorption peaks at 254 and 350 nm, at the same peak position as folic acid, which demonstrated the successful conjugation of folic acid with alginate. The reason for the bathochromic shift of folic acid in the AF NPs is the polarity change of the solvent. In the UV Spectroscopy measurement, the solvent of AF was DI water and the solvent of folic acid was DMSO.

Ionic gel formation is one of the most important properties of alginate. The gelation effect is caused by divalent ions which allow a high degree of coordination with guluronate blocks, described as the egg box model [36]. Generally, calcium chloride is used as a cross-linking agent for alginate gels due to its prevalence in vivo. Gelation was performed with calcium chloride to verify whether AF shared the characteristic properties of alginate. When 1.5 wt% of calcium chloride was added to 1 wt% of both alginate and AF, gelation occurred within a short time (within 5 s) (Figure 4).

### 3.2. Stability and Physical Characterization of AF NPs

Different NPs were prepared by varying the AF concentration (0.5–1 wt%) and co-surfactant ratios. The characterization results for the various types of NPs are shown in Appendix A. AF NPs were fabricated by the W/O emulsification technique. The NPs were stored in PBS and the mean size of NPs was characterized by dynamic light scattering (DLS). As seen in Appendix A, the smallest size of NP was approximately 45 nm, which could efficiently penetrate tumors and result in a higher internalization rate [37,38]. ζ potential is an important indicator of the stability of NPs and is caused by net electrical charge on the nanoparticle surface in solution. All NPs had a negative average surface charge due to the carboxyl group of the alginate, with an absolute value of 20 or more. The largest absolute value of ζ potential, −29.3 ± 0.1 mV, was observed for the smallest nanoparticle (Appendix A). These results revealed that NPs formed a stable suspension. AF NPs were stable over a period of three months as evidenced by the measurement of the size distribution (Figure 5B). The AF nanoparticle size distribution diagrams are shown in Appendix A. The TEM micrograph and size distribution of AF nanoparticles are shown in Figure 5A. The TEM image clearly shows that the NPs have spherical morphology and are monodisperse, without apparent aggregation. The size distribution of NPs was obtained by analyzing the diameter of 230 particles, and the average size was found to be 25 nm.

To confirm that the negatively charged AF NPs were affected by proteins in the cell medium, the size distribution of AF NPs stored in 4 °C PBS and AF NPs cultured in the cell medium at 37 °C for 24 h was compared (Appendix A). AF NPs cultured in the medium for 24 h were observed to have effect on proteins as the size increased by 2 nm from 42 to 44 nm in diameter.

The 5ALA loading capacity (LC) and encapsulation efficiency (EE) of the various AF NPs are shown in Appendix A. LC increased with increasing AF concentration, and EE also increased with 1 wt% AF yielding an EE four times greater than that of 0.5 wt%. This was due to an increase in the number of ionic bonds between the alginate nanocarrier and the amphoteric ionic polysaccharide of 5ALA [39]. Therefore, as the AF concentration increased, a higher concentration of 5ALA could be encapsulated, thereby increasing the EE. Based on these results, NP4 was used for in vitro studies.

### 3.3. Release Profiles of 5ALA

The release kinetics of 5ALA from AF NPs were investigated at pH 5.0 and pH 7.4. As shown in Figure 6, no more than 30% of 5ALA was released over the experimental period at pH 7.4. These results indicate that the 5ALA contained in AF NPs remained stable under physiological conditions. However, more than 80% of the encapsulated 5ALA is released after 80 h at pH 5, with exponential release observed between 12–24 h. This exponential release was due to the deprotonation and decomposition of the alginate carboxylic acid group at acidic pH [10,13,14]. Therefore, acid-catalyzed hydrolysis of alginate occurs in acidic environments, and AF NPs decompose to release the drug. These results suggest the potential for use in endoscopy as a target for gastrointestinal tract cancer. Even if 5ALA is released at pH (~7.4) in vivo, it will not exhibit a signal-to-noise ratio (SNR) because the selectivity of normal and cancer cells of 5ALA is so good.

### 3.4. Quantitative Measurement of Generated PpIX and Cytotoxicity

PpIX accumulation in cells was quantitatively measured. To express the PpIX concentration proportional to the number of cells, the fluorescence intensity was normalized to the amount of protein extracted from the cells measured by BCA assay. The intracellular protein concentration calculated by the BCA assay is shown in Appendix A. The fluorescence measurements of the different types of NPs, dispersed in DI water or PBS and incubated with cancer cells for 24 h, is shown in Figure 5C. The fluorescence intensity was higher for the samples dispersed in PBS than those in DI water. Sample NP4 showed the strongest fluorescence intensity, which was proportional to the EE value of these particles. Therefore, NP4 was used for all in vitro tests. The change in fluorescence intensity over time in the cancer cell lines in comparison with normal HFB cells is shown in Figure 7E. No fluorescence was observed in the normal cells, whereas the fluorescence intensity showed a gradual increase in cancer cells up to 12 h, and then rapidly increased until 24 h. FR-α is expressed at similar levels in MCF7 and A549, while it is abundantly expressed in SKOV3. Therefore, the fluorescence intensity of A549 and MCF-7 for 24 h was similar, and that of SKOV3 was higher than that of the two cell lines [40].

The cytotoxicity of AF NPs was investigated through CCK-8. The biocompatibility of the NPs was demonstrated by expressing the cell viability against the NP concentration and incubation time (Appendix A). Folic acid and 5ALA are normally present in the body, and alginate is also a natural product, and is a clinically approved substance. Therefore, AF NPs did not display high levels of cytotoxicity even at the highest concentration for 24 h [24,28,41,42]. 

### 3.5. Cellular Internalization

Fluorescence imaging was carried out using a fluorescence microscope to observe whether the AF NPs were up-taken to the cells and to show the accumulation of PpIX. Fluorescence images were measured after an appropriate incubation period (12 h, 24 h) for alginate decomposition to occur in the cells. In the normal HFB cell line, no fluorescence was observed after NP treatment (Figure 7A). However, fluorescence began to appear in the three cancer cell lines 12 h after incubation with NPs (Figure 7B–D). In Appendix A, there was an effect of protein on AF NPs, but as a result, the recognition by membrane epitope was not a significant problem in targeting. When the AF NPs were up-taken from the cells, the selectivity was good without a significant problem in targeting.

To demonstrate the role of folic acid in the internalization of AF NPs, NPs were prepared with alginate only and fluorescence imaging was performed. After incubation with alginate NPs for 24 h, no fluorescence was observed in normal cells and fluorescence was observed in cancer cells (Figure 7A–D). This suggests that the folic acid-modified NPs bind to folic acid receptors specifically overexpressed in cancer cells, thereby efficiently delivering 5ALA to cancer cells. AF NPs were internalized by cells through folate receptor-mediated endocytosis and degraded at the lysosomal/endosomal pH. Subsequently, the 5ALA released from the NPs was converted to PpIX through the heme synthesis pathway, which led to its accumulation in cancer cells and resulted in fluorescence. In normal cells, the small amount of 5ALA released from endocytosed NPs can be fully converted to heme without resulting in accumulation of PpIX, therefore no fluorescence is visualized.

## 4. Conclusions

In this study, NPs containing a fluorescent precursor specific for cancer cells and with longer circulation times were successfully produced. Folic acid was conjugated with alginate, used to synthesize NPs that were loaded with 5ALA, and specifically incorporated into cancer cells through receptor-mediated endocytosis. The release of 5ALA from the NPs was pH sensitive and was caused by the hydrolysis of alginate in the acidic lysosomal environment of cancer cells. The NPs were not degraded by enzymes or other external factors before reaching the target site and were stably delivered to the cancer cells. In addition, the NPs were characterized by cytotoxicity, stability, and uniformity with favorable results. The outcomes of the present study are significant, as they delineate a strategy to prevent drug leakage from NPs during circulation and increase drug distribution to the target tissue due to specific uptake of NPs by cancer cells. Since the intracellular pH is complex, additional research is needed so that it can be delivered to the target site without being decomposed for various in vivo pHs. The supported photosensitizer (5ALA) can be extended to an animal model to study whether the PDT effects are present in the cancerous lesion site. Anti-cancer drugs such as doxorubicin or paclitaxel are carried on the developed NPs, which can be applied to cancer treatment as an anticancer drug carrier.

## Figures and Tables

**Figure 1 pharmaceutics-12-00537-f001:**
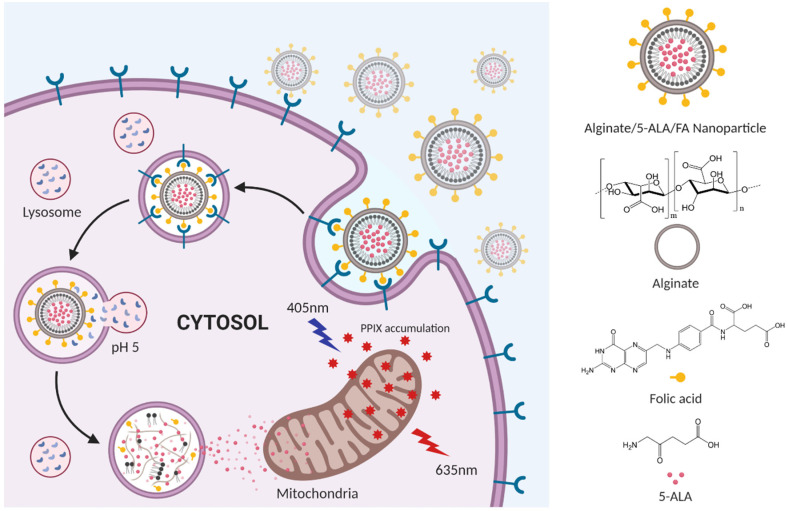
Schematic illustration of 5-aminolevulinic acid hydrochloride (5ALA) delivery and release to cancer cells by alginate and folic acid modified nanoparticles.

**Figure 2 pharmaceutics-12-00537-f002:**
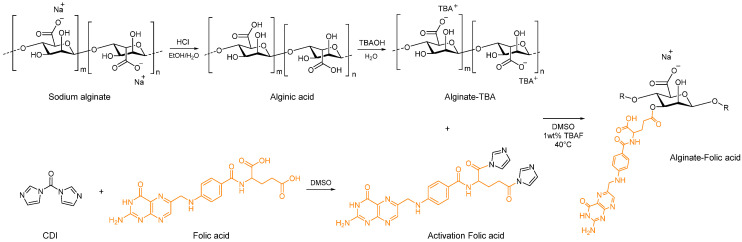
Schematic diagram representing alginate conjugation to folic acid.

**Figure 3 pharmaceutics-12-00537-f003:**
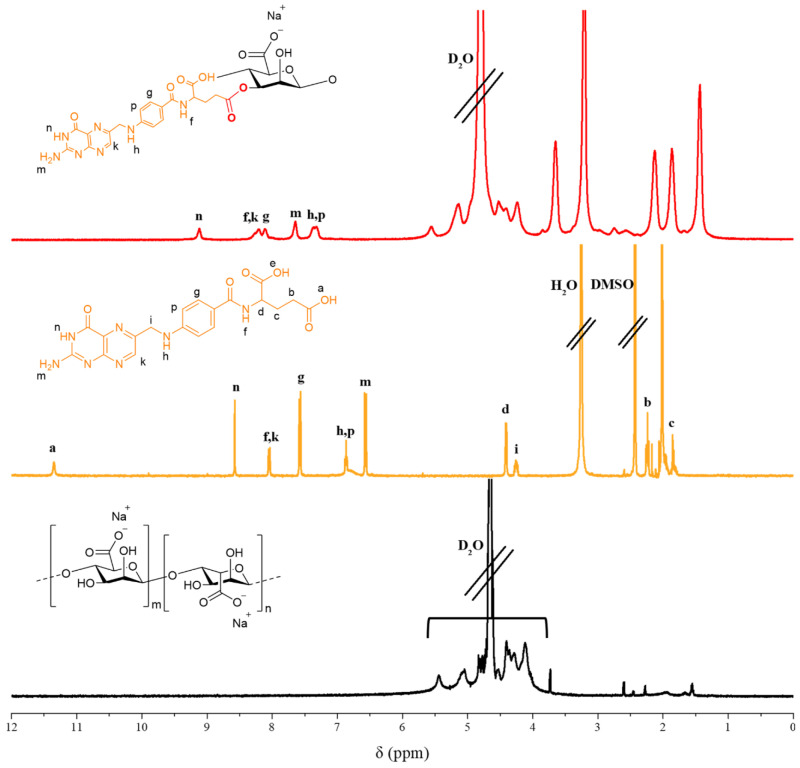
^1^H NMR spectrum of alginate-conjugated folic acid (AF), folic acid and alginate.

**Figure 4 pharmaceutics-12-00537-f004:**
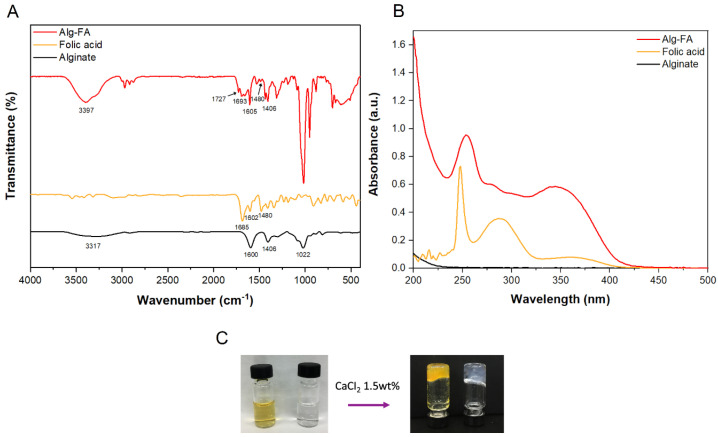
(**A**) FT-IR spectra and (**B**) UV-Vis spectroscopy of AF, folic acid and alginate. (**C**) Gelation behavior of AF and alginate. AF (left) and alginate (right) were dissolved in deionized (DI) water at 1 wt% and treated with 1.5 wt% CaCl_2_ followed by visual inspection of gelation.

**Figure 5 pharmaceutics-12-00537-f005:**
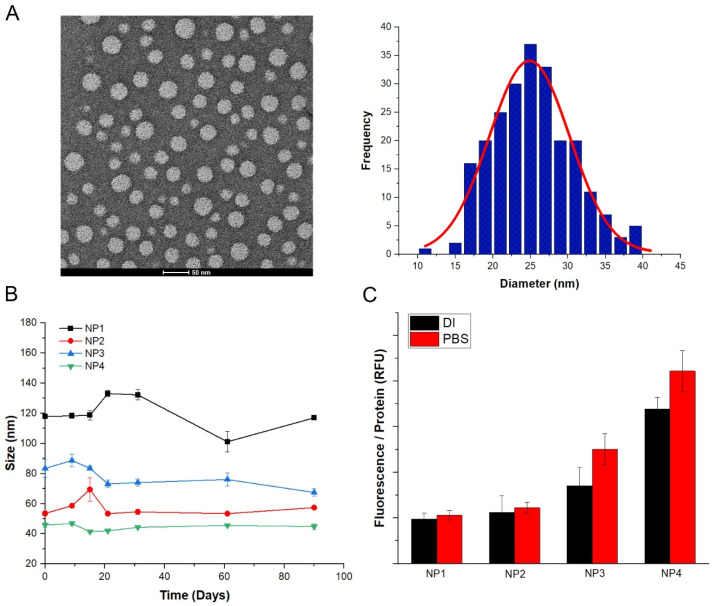
Physicochemical characterization of nanoparticles (NPs). (**A**) Transmission electron microscopy (TEM) image showing the morphology of NP4. The size distribution was calculated by measuring the diameters of 230 NPs. Scale bar = 50 nm. (**B**) Stability of NPs over time. The four different NP suspensions were stored at 4 °C in PBS for three months, and the size change was measured by dynamic light scattering (DLS). (**C**) Quantitative fluorescence measurement of NPs. The NPs were dispersed in either deionized (DI) water or PBS and incubated for 24 h with human ovarian adenocarcinoma (SKOV-3) cells, and subsequently fluorescence measurements were acquired. The relative fluorescence unit (RFU) of PpIX was normalized to the amount of intracellular protein measured by bicinchoninic acid (BCA) assay. All data are presented as mean ± SEM.

**Figure 6 pharmaceutics-12-00537-f006:**
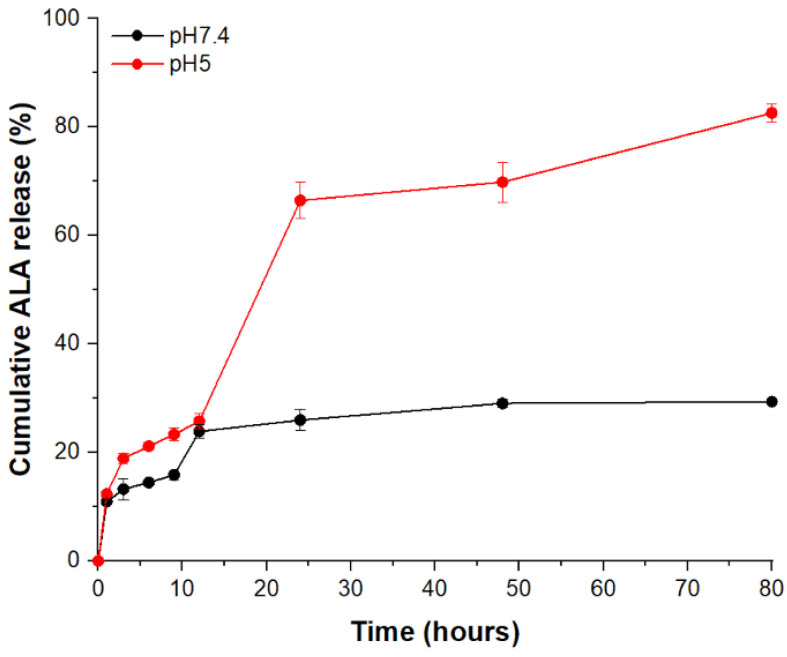
In vitro cumulative release kinetics of 5ALA from AF NPs at 37 °C. Evaluation was performed at pH 7.4 and pH 5. The cumulative 5ALA release was measured by 2,4,6-trinitrobenzene sulfonic acid (TNBSA) assay at 435 nm.

**Figure 7 pharmaceutics-12-00537-f007:**
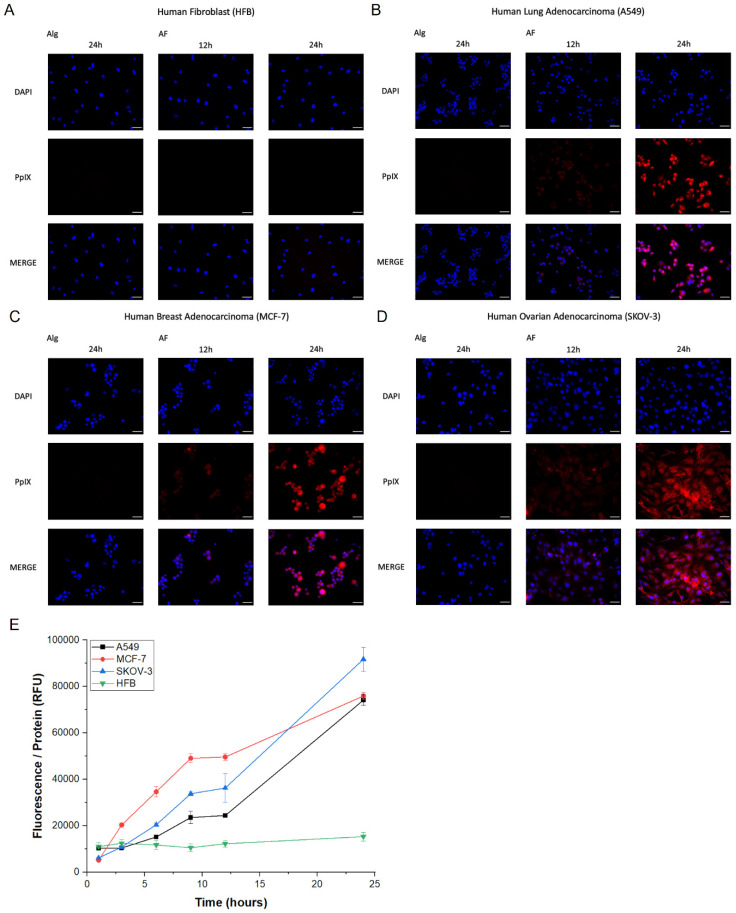
Fluorescence imaging of intracellular PpIX generation in (**A**) human fibroblast (HFB), (**B**) human lung adenocarcinoma (A549), (**C**) human breast adenocarcinoma (MCF-7), and (**D**) human ovarian adenocarcinoma (SKOV-3) cell lines. Representative fluorescence micrographs of normal cells (**A**) and tumor cell lines (**B**–**D**) show cells after 12 and 24 h incubation with AF NPs. Nuclei were stained with 4′,6-diamidino-2-phenylindole (DAPI) (blue), PpIX is shown in red. To characterize the role of folic acid, alginate NPs were incubated with cells for 24 h, and the fluorescence micrographs obtained are compared with those of AF NPs. Scale bar = 50 μm. (**E**) Quantitative fluorescence measurements of PpIX generated by AF NPs in three cancer cell lines (A549, MCF-7, and SKOV-3) and one non-cancerous cell line (HFB). All data are presented as the mean ± SEM.

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
