# Peer review of "pH-Sensitive Folic Acid Conjugated Alginate Nanoparticle for Induction of Cancer-Specific Fluorescence Imaging"

_pharmaceutics, 2020, doi:10.3390/pharmaceutics12060537_

Round 1

Reviewer 1 Report

Referee Report

Title: pH-sensitive folic acid conjugated alginate nanoparticle for induction of cancer-specific fluorescence imaging.

Manuscript ID: pharmaceutics-819413

By Lee et al

Submitted to Pharmaceutics (ISSN 1999-4923)

Comment

This work suggested a pH-sensitive folic acid conjugated alginate NPs as the cell fluorescence imaging agent. By controlling the pH of the cell environment, the NPs can release ALA to the target site. This will prevent the NPs degraded by enzymes or other factors for the drug delivery in the cellular. This work is quite comprehensive and the experimental works are quite thorough. I only have some concerns:

  1. The authors selected some cancer cell lines. However, they did not show how the NPs can highlight the tumour cells with the surrounding healthy cells unaffected (i.e. a tumour environment). This will require a small-animal model. In this work, it can only prove that the NPs can be used as a fluorescence imaging agent, but it cannot prove the NPs can function in the cancer imaging to identify the tumour from living tissues in the patient. This should be the goal to develop these NPs.

  1. There are many works related to controlling the pH level of the cellular environment in drug delivery using NPs. The authors should stress how and why their suggested NPs are better than others in biomedical application.

  1. Preclinical model on the NPs is desired. It is not required here but at least the authors should discuss what they are going to do in the manuscript as future work.

  1. The authors should mention if there is any potential application of their suggested NPs in cancer therapy?

  1. Figure 3: There is no label for the NMR spectrum of alginate.

  1. Figure 7 is very important in this work. However, the subfigures are too small to view. They should make bigger for better illustration.

Reviewer 2 Report

  1. Please include the complete details of alginate used. e.g. molecular weight 
  2.  The authors should determine the molecular weight of formed conjugate and include in the revised manuscript.
  3. An explanation is needed for esterification by at particular carboxylic acid function of folic acid by alginate. Why esterification did not occur at another adjacent carboxylic acid and why esterification did not take place at both carboxylic acid functions?
  4. Short term stability study on developed nanoparticles needs to be performed. 

Reviewer 3 Report

The paper described a pH-sensitive NP for cancer targeted fluorescence imaging. I recommend the authors to revise the points as below:

  1. Figure 7E: Both MCF-7 and SKOV-e cells are folic acid (FR)-positive, while A549 is FR-negative. This might partially explain the difference of the three cell lines within 12 h in Figure 7E. But why A549 and MCF-7 were similar at 24 h?
  2. Figure 6 showed the NPs were pH-sensitive in PBS, however, the intracellular pH is complicated. It is better to culture cells in different pH buffer, then observe whether the release of ALA is different for at least one cell line.
  3. Page 5, section 2.9: The excitation laser and the filters should be mentioned. Line 176: “preincubated at 37℃.” The preincubation time should be added.

Numerous grammatical errors need to be addressed throughout the manuscript, to name just a few:

  1. Page 5,line 175: 0.05x106 cells should be 5x104

     2. Page 5,  lines 186-187:  FT-IT should be FT-IR.

     3.  Line 275: Figure 6 should be changed to Figure 7.

Reviewer 4 Report

Lee et al. present a nanoparticle(NP) based delivery scheme for an FDA approved PDT drug, 5-ALA. The nanoparticle conjugation, incorporation, release pathways were validated. NPs were also tested for toxicity and other non-functional characteristics, including chemical stability and physical features. The research is well-designed for NP testing. The authors present a modification of nanoparticle-based drug delivery systems. They developed alginate-conjugated folic acid nanoparticles (AF NPs) with dimensions small enough to be internalized by cancer cells, and its disintegration is pH-dependent. The AFNP delivery was demonstrated by enclosing 5-aminolevulinic acid (5ALA), a precursor to protoporphyrin IX (PpIX), that is detected by fluorescence microscopy.

  • My main question is, how are authors distinguishing the internal release of 5ALA using just timelapse of PPIX intensity? Cancer cells are known to perturb the buffer pH, and hence PPIX release could also happen in the media instead of internalization. Could any supplementary evidence be supported for NP internalization?

Below are my comments/ suggestions.

* I believe the commonly used acronym for 5-aminolevulinic acid hydrochloride is 5ALA. Consider using 5ALA instead of ALA.

* Provide a brief explanation of the process demonstrated in Fig 1 schematics (either in the text or in figure caption)

* In the introduction, motivate why the developed NP were made to degrade in an acidic environment (explaining lysosomal pH, cancer cell pH and so on)

* Please cite previous works that have developed NP enclosing 5-aminolevulinic acid in the introduction section.

* Why was cytotoxicity tested for only 24 hours? This is very low for even for cell lines with doubling rates of 24hours (all the listed cells have lower rates). There could be effects at longer times. Justify the short-time toxicity-tests or cite others who used similar fashioned tests.

* For characterizing internalization of NPs (section 3.5) by fluorescence imaging, how many repeats were performed for each cell line? How have regions imaged per condition to obtain the data and standard deviation in Fig 7E?

* Figure 7 is too small in size and hard to validate the internalization. Please provide larger imaged or zoomed-in insets. What is the acronym RFU in the Figure 7.E? 

* The microscopy section (2.9) for figure 7 doesn't explain the filters used, imaging parameters, etc.? It would be useful to list them. Especially if the laser powers were maintained the same and the scope of the LUT used, etc.

* In the discussion section, discuss the limitations of the developed NP-delivery scheme.

Reviewer 5 Report

Report attached.

Round 2

Reviewer 1 Report

I am satisfied with the responses from the authors and their corrections regarding my comments. I have no further question in this submission.

Reviewer 3 Report

The revised manuscript is recommended to publish.

Reviewer 4 Report

The manuscript has improved with the revision. The authors have addressed all the listed concerns in my last review, convincingly. I find the article promising for cancer diagnostics and nano-particle based photo-sensitizer therapy. The implementation is interesting and easy to reproduce in a chemical lab. I appreciate the authors sustaining efforts in this research area.